# Deciphering the Hazardous Effects of AFB1 and T-2 Toxins: Unveiling Toxicity and Oxidative Stress Mechanisms in PK15 Cells and Mouse Kidneys

**DOI:** 10.3390/toxins15080503

**Published:** 2023-08-14

**Authors:** Shuai Xiao, Yingxin Wu, Suisui Gao, Mingxia Zhou, Zhiwei Liu, Qianbo Xiong, Lihuang Jiang, Guoxiang Yuan, Linfeng Li, Lingchen Yang

**Affiliations:** 1College of Veterinary Medicine, Hunan Agricultural University, No. 1 Nongda Road, Furong District, Changsha 410128, China; shuaixiao@stu.hunau.edu.cn (S.X.); wuyingxin@ipm-gba.org.cn (Y.W.); gaoss@stu.hunau.edu.cn (S.G.); mingxiazhou@stu.hunau.edu.cn (M.Z.); bobx972@stu.hunau.edu.cn (Q.X.); jianglihuang@stu.hunau.edu.cn (L.J.); yuanguoxiang@stu.hunau.edu.cn (G.Y.); 13407319966@stu.hunau.edu.cn (L.L.); 2Wuhan Animal Disease Control Center, No. 170, Erqi Road, Jiang’an District, Wuhan 430014, China; liuzhiwei0516@163.com

**Keywords:** AFB_1_, T-2, combined effect, kidney damage, oxidative stress

## Abstract

In China, animal feeds are frequently contaminated with a range of mycotoxins, with Aflatoxin B1 (AFB1) and T-2 toxin (T-2) being two highly toxic mycotoxins. This study investigates the combined nephrotoxicity of AFB1 and T-2 on PK15 cells and murine renal tissues and their related oxidative stress mechanisms. PK15 cells were treated with the respective toxin concentrations for 24 h, and oxidative stress-related indicators were assessed. The results showed that the combination of AFB1 and T-2 led to more severe cellular damage and oxidative stress compared to exposure to the individual toxins (*p* < 0.05). In the in vivo study, pathological examination revealed that the kidney tissue of mice exposed to the combined toxins showed signs of glomerular atrophy. The contents of oxidative stress-related indicators were significantly increased in the kidney tissue (*p* < 0.05). These findings suggest that the combined toxins cause significant oxidative damage to mouse kidneys. The study highlights the importance of considering the combined effects of mycotoxins in animal feed, particularly AFB1 and T-2, which can lead to severe nephrotoxicity and oxidative stress in PK15 cells and mouse kidneys. The findings have important implications for animal feed safety and regulatory policy.

## 1. Introduction

Aflatoxin B1 (AFB1) and T-2 toxin (T-2) are potent mycotoxins that pose a significant threat to human and livestock health. Aflatoxins (AFs) are furocoumarin derivatives with similar chemical structures [1], produced primarily by two types of fungal microorganisms, *Aspergillus flavus strain* and *Aspergillus niger* (AFB1, AFB2, AFG1, and AFG2) [2,3]. T-2 is a type of trichothecene mycotoxin that is widely distributed across the world and can cause sublethal or lethal toxicosis, ultimately impacting the growth of humans and animals [4]. It is important to investigate the combined toxicity of mycotoxins on organisms in real-life settings, as research has indicated that the toxicity of naturally mold-containing feed is much greater than that of purified mycotoxins [5].

AFB1 exhibits the most potent mutagenic and carcinogenic activity among aflatoxins. For this reason, AFB1 is recognized as a human group 1 carcinogen by the International Agency of Research on Cancer [6]. As a potent carcinogen, AFB1 reacts mainly with liver DNA and serum albumin in a dose-dependent manner [7]. Previous studies have established that the liver, which plays a crucial role in the detoxification and metabolism of drugs and toxins within the body, is the primary target organ of AFB1 [8,9]. AFB1 exposure can cause marked changes in enzymatic or non-enzymatic molecules of oxidative and antioxidant signaling, resulting in severe impact on liver function in animals [10]. AFB1 also has strong immunosuppressive toxicity, significantly reducing the number and function of lymphocytes, macrophages, and natural killer cells, with cell-mediated immunity and phagocytic cell functions being the primary affected components [11,12]. In addition, AFB1 exposure can lead to growth stunting and malnutrition, causing numerous instances of poisoning in various countries and regions.

T-2 exposure can lead to weight loss, anorexia, vomiting, diarrhea, gastrointestinal bleeding, and necrosis in humans or animals, resulting from varying degrees of toxicity through oral, dermal, or air exposures [13,14,15,16]. Furthermore, T-2 can harm multiple systems, such as the digestive, nervous, immune, and reproductive systems of animals, leading to non-fatal and fatal toxicities [17]. For example, T-2 exposure can cause alimentary toxic aleukia (ATA) in humans [18,19,20]. Therefore, understanding the toxicity of mycotoxin exposure, both individually and combined, is crucial for mitigating the health risks associated with these toxins.

Research on the toxic effects of combined mycotoxins has gained significant attention in recent years. Studies have demonstrated the synergistic effects of mycotoxins, including AFB1 and T-2, on the health of animals and humans. For example, the synergistic effects of AFB1 and T-2 were observed in broiler chickens [21], while dose-dependent and time-dependent effects of Fumonisin B1 (FB1), Deoxynivalenol (DON), T-2, and Zearalenone (ZEA) on cells were found [22]. The combined effect of feeding AFB1 and FB1 to mice resulted in the higher production of interleukin-10 by splenic monopoiesis and lower interleukin-2 production compared to the single toxin effect, suggesting that the combined effect affects immune function [23]. However, there is a lack of studies on the in vivo and in vitro effects of the combination of AFB1 and T-2 in animals.

Therefore, the purpose of this study is to investigate the toxicity and oxidative stress of AFB1 and T-2 on porcine kidney-15 cells (PK15 cells) and mouse kidneys. In vitro experiments were conducted to understand the toxic effects of the combined action of AFB1 and T-2 on PK15 cells. Specifically, we incubated PK15 cells with AFB1 and T-2 and measured various parameters, including cellular activity, GSH-Px activity, GSH and MDA content of the antioxidant enzyme system, intracellular reactive oxygen species (ROS) content, cellular pathological changes, and changes in mRNA expression levels related to oxidative stress. In vivo experiments were also conducted, where AFB1 and T-2 were orally administered to mice alone or in combination. The effects of the combined action of the toxins on renal function in mice were determined by measuring changes in food intake, body weight, biochemical indices related to renal tissue, and renal histopathology. Additionally, changes in GSH-Px activity, GSH, MDA, and ROS content of kidney tissue, as well as changes in mRNA expression levels related to oxidative stress were measured. The primary objective of this study is to elucidate the synergistic toxic effects of AFB1 and T-2 on PK15 cells and murine renal tissues, while concurrently exploring the underlying mechanisms involving oxidative stress pathways.

## 2. Results

### 2.1. Cellular Activity and IC50

The cell activity of PK15 cells treated with different concentrations of toxins was measured to evaluate their cytotoxicity. As illustrated in Figure 1A, the cell activity of the PK15 cells decreased appreciably (*p* < 0.05) at AFB1 concentrations above 3125 nM in comparison to the control group (NC). The lowest cell activity was observed at the 50,000 nM concentration, but no significant difference was observed in cell activity between the 25 and 50,000 nM concentrations (*p* > 0.05). Similarly, Figure 1B depicts a significant reduction in cell activity at T-2 concentrations above 0.001 nM (*p* < 0.05). There was a significant decrease in cell activity in all cases as the T-2 concentration increased to 0.1 nM. Cell activity was markedly inhibited when T-2 concentrations rose from 1 to 10 nM. However, no notable difference in cell activity was observed between the 0.1 and 1 nM T-2 concentrations (*p* > 0.05).

To further assess the cytotoxicity of AFB1 and T-2, their IC50 values were calculated as 30,060 nM and 1.179 nM, respectively, by probit analysis, as shown in Table 1. IC50 approximations of 30,000 nM and 1.2 nM were subsequently employed for further tests.

### 2.2. Cellular Toxicity Induced by the Synergistic Action of AFB1 and T-2

The study evaluated the combined toxicity of two mycotoxins, AFB1 and T-2, on PK15 cells. The two toxins were mixed at various concentrations and co-incubated with normal cells for 24 h, as shown in Table 2. The cellular activity was determined using the MTT assay, and the actual IC50 value of the binary mixture of toxins was 23,273 nM, which was calculated using probit analysis. The concentration of AFB1 in the mixture was 23,272 nM, and the concentration of T-2 was 1 nM.

In Figure 1C the combined action of AFB1 and T-2 in PK15 cells produced a dose–response relationship, with the toxin-treated group achieving a minimum of 40.4% and maximum of 83.48% cell survival. The combined toxicity index K was calculated as the mean sum of IC50 of each toxin divided by the measured IC50 value of the combined toxin, which was 1.29. The formula showed that the combined action of AFB1 and T-2 produced a synergistic effect.

The AFB1 group and T-2 group were denoted as the groups in which AFB1 and T-2 acted on PK15 cells alone, respectively. The IC50 values of the toxins derived from the AFB1 and T-2 groups were simply summed, and the calculated average value was used as an estimated IC50 concentration of 15,000.6 nM after the combined action of the two toxins, expressed as the ATG group. The actual IC50 concentration after combined action was 23,273 nM, expressed as the ATS group.

### 2.3. Cellular Morphological Evaluation in an In Vitro Model Exposed to AFB1 and T-2 Alone and in Combination

In Figure 2 the morphological changes observed in the AFB1, ATG, and ATS groups included reduced cell adhesion and increased intercellular spaces, which were consistent with the decrease in cell viability observed in the MTT assay. In Figure 3 the HE-stained images confirmed these findings, with the AFB1, ATG, and ATS groups showing significant changes in cell morphology compared to the NC group. The T-2 group, on the other hand, did not show significant differences in cell morphology compared to the NC group, indicating that the dose of T-2 used in the study had minimal effects on cell activity at the 24 h time point. Overall, the morphological changes observed in the study were consistent with the toxic effects of AFB1 and T-2 observed in the MTT assay.

### 2.4. Quantification of ROS in Cells Exposed to Toxins

The study conducted ROS measurements in both toxin alone and combined toxin groups, and the results are illustrated in Figure 4. The ROS-positive cell rate in all groups showed a remarkable increase compared to the NC group (*p* < 0.05). However, the AFB1, ATG, and ATS groups had markedly more ROS-positive cells than the T-2 group (*p* < 0.05).

### 2.5. Assessment of Intracellular MDA, GSH, and GSH-Px Levels in Toxin-Treated Cells

Figure 5 presents the histograms of MDA concentration, GSH concentration, and GSH-Px activity following the 24 h of toxin treatment. The MDA concentrations were noticeably higher in all toxin groups compared to the NC group (*p* < 0.05). Notably, no significant difference was observed between the AFB1, ATG, and ATS groups terms of MDA concentrations (*p* > 0.05).

In terms of GSH levels, the T-2 group showed a significant increase compared to the NC group (*p* < 0.05), while the ATG and ATS groups exhibited a significant decrease in GSH levels (*p* < 0.05). Conversely, the AFB1 group showed no significant difference in GSH levels compared to the NC group (*p* > 0.05).

Regarding GSH-Px activity, a significant increase was observed in the AFB1, ATG, and ATS groups compared to the NC group (*p* < 0.05). In contrast, no significant difference in GSH-Px activity was found in the T-2 group (*p* > 0.05). Moreover, GSH-Px activity was considerably higher in the ATG and ATS groups compared to the AFB1 group (*p* < 0.05).

### 2.6. Evaluation of Oxidative Stress-Related mRNA Expression in Cells

The study investigated the relative expression of oxidative stress-related mRNAs, including GSH-Px, Nrf2, Hmox1, NQO1, and Keap1, in response to toxin treatments. The results are presented in Figure 6. The analysis showed that the relative expression of GSH-Px mRNA was significantly elevated in the ATS group compared to the NC group (*p* < 0.05). However, there were no significant differences in the relative expression levels of GSH-Px between the AFB1, T-2, and ATG groups (*p* > 0.05), although an increasing trend was observed in AFB1 and ATG, while T-2 showed a decreasing trend.

The results indicated that the relative expression of Nrf2 was significantly elevated in the AFB1 group compared to the NC group (*p* < 0.05), while the relative expression of Nrf2 was significantly decreased in the T-2, ATG, and ATS groups (*p* < 0.05). There were no significant differences in Nrf2 relative expression levels between the ATG and ATS groups (*p* > 0.05).

The analysis also showed that the relative expression of Hmox1 mRNA was materially increased in the AFB1, ATG, and ATS groups compared to the NC group (*p* < 0.05). However, there were no significant differences in Hmox1 relative expression levels in the T-2 group (*p* > 0.05).

Regarding the NQO1 mRNA relative expression, the results indicated that NQO1 relative expression was considerably elevated in all study groups compared to the NC group (*p* < 0.05).

Finally, the results of the Keap1 mRNA relative expression analysis showed that it was noticeably increased in the AFB1, ATG, and ATS groups compared to the NC group (*p* < 0.05), while there was no significant difference in the T-2 group (*p* > 0.05).

### 2.7. Body Weight, Weight Gain, and Food Intake

The body weight, weight gain, and food intake of the mice in each group are presented in Table 3. The terminal body weight of the mice in the AFB1 group did not differ remarkably from that of the NC group (*p* > 0.05). However, the terminal body weight of the mice in the T-2 and AT groups was notably lower than that of the NC group (*p* < 0.05). Additionally, the average daily food intake of the AT group was significantly lower than that of the other groups (*p* < 0.05).

### 2.8. Serum Biochemistry Analysis Parameters

The results of serum ALT, AST, ALP, BUN, TP, TC, and TG measurements are presented in Figure 7. Among these parameters, the values of ALT, AST, BUN, and TC did not show significant differences (*p* > 0.05) among the toxin-treated groups. Conversely, compared to the NC group, the values of ALP were materially lower in all toxin-treated groups (*p* < 0.05). The TP values showed a significant decrease in the T-2 and AT groups (*p* < 0.05), while the difference was not significant in the AFB1 group. Additionally, the values of TG showed an increasing trend in the AFB1 and T-2 groups and a decreasing trend in the AT group, but these changes were not statistically significant (*p* > 0.05).

### 2.9. Histopathological Analysis of Kidneys

The histopathological analysis of the kidneys was performed using HE staining of paraffin sections, and the results are presented in Figure 8. The glomeruli and tubules in the NC group showed normal morphology and no pathological issues. The tissue structure and morphology of the AFB1 group were similar to those of the NC group. However, the T-2 and AT groups exhibited glomerular atrophy, interstitial edema, dilated tubules, and blurred tubular structures with inflammatory cell infiltration.

### 2.10. Quantification of ROS in Kidneys

The ROS assay was performed in kidney tissue and the results are presented in Figure 9. In the kidney tissue, the level of ROS was markedly reduced in the AFB1 group compared to the NC group (*p* < 0.05), while the level of ROS was significantly elevated in the T-2 and AT groups (*p* > 0.05).

### 2.11. Assessment of Intracellular MDA, GSH, and GSH-Px Levels in Kidneys

The results of MDA, GSH, and GSH-Px measurements in kidney tissue are shown in Figure 10. In kidney MDA, only the AT group showed a notable increase compared to the NC group (*p* < 0.05). GSH levels were significantly amplified (*p* < 0.05). GSH-Px activity was only appreciably diminished in the AFB1 group compared to the control (*p* < 0.05); the difference between the T-2 and AT groups was not statistically significant (*p* > 0.05).

### 2.12. Evaluation of Oxidative Stress-Related mRNA Expression in Kidneys

The expression of mRNA associated with oxidative stress in renal tissues is presented in Figure 11. GSH-Px mRNA and Keap1 mRNA expression was markedly decreased in the toxin group compared to the NC group (*p* < 0.05), Nrf2 mRNA expression was remarkably augmented in the AT group compared to the NC group (*p* < 0.05), and Hmox1 mRNA expression was notably reduced in the AT group compared to the NC group (*p* < 0.05). The expression level of Hmox1 mRNA was notably decreased in the AT group compared to the NC group (*p* < 0.05).

## 3. Discussion

Mycotoxins are fungal secondary metabolites that persistently contaminate food and feed, posing a significant threat to human and animal health [24]. Previous studies have indicated that mixed contamination of food and animal feed with fungal toxins is widespread [25]. Amongst the various mycotoxins, AFB1 and T-2 are considered to be the most potent in terms of their respective toxin species [4,26]. As a result, extensive research has been conducted on their cytotoxicity, oxidative stress, and apoptosis. Notably, interactions between AFB1 and T-2 have been observed in cells, producing a superimposed effect [27]. Despite this, the combined toxicity of AFB1 and T-2 in PK15 cells and in KM mice has received limited attention.

To address this knowledge gap, the objective of this study was to investigate the combined toxicity of AFB1 and T-2 in vitro and in vivo and the associated oxidative stress mechanisms. In vitro, PK15 cells were treated with a combination of AFB1 and T-2 to evaluate their toxicity and oxidative stress responses. The findings of this study revealed that AFB1 and T-2, alone or in combination, produced time- and dosage-dependent increasingly toxic effects on PK15 cells, which is similar to the effects of T-2 toxin and its metabolites observed on porcine Leydig cells [28]. Our initial findings revealed that, at equivalent doses, cell viability in the T-2 group was significantly lower than in the AFB1 group (Figure 1A,B), indicating that T-2 had a more pronounced effect on cell viability at the same concentration.

The combined action of mycotoxins may be additive, synergistic, or antagonistic. Based on the analysis of the combined toxicity index K, this study concludes that AFB1 and T-2 have a synergistic effect on PK15 cells with IC50 after 24 h. This is consistent with the finding by Lili Hou et al., that the combination of AFB1 and OTA exacerbates immunotoxicity [29]. Wu Chenqing also found that combined treatment of HepG2 cells with AFM1 + OTA caused metabolic abnormalities and more severe cytotoxicity than mycotoxin treatment alone [30]. In Vero kidney cells, AFB1 and OTA not only showed an additive cytotoxic effect but also synergism to promote genotoxicity with increased DNA fragmentation [31]. There experiments indicate that the binary and ternary combinations of T-2, HT-2, and NEO displayed synergistic toxicity in porcine Leydig cells [28]. However, it should be noted that differences in cell model, toxin concentration, and incubation time may affect the effects of toxin combinations. For instance, Klaric et al. found that the combination of Beauvericin, FB1, and OTA on PK15 cells showed mainly additive effects in binary and ternary mixtures [32,33].

Furthermore, the sensitivity to mycotoxins varies from cell to cell. In this study, the IC50 of AFB1 and T-2 to PK15 cells was determined to be 30,060 nM and 1.179 nM, respectively, and the IC50 after the combination of the two toxins was determined to be 23,273 nM. Cellular activity decreased with the increasing concentrations of mixed toxins over a range. The exception is when the toxin concentration is low or too high. At concentrations of 3750.15 and 7500.3 nM, the toxin dosage was low, allowing the cells to withstand the toxin to some degree. Consequently, there was a reduction in cell viability at these two concentrations, but no significant difference between them. Cell viability decreased as the toxin concentration increased, but was slightly higher at 30,001.2 nM than at 26,251.05 nM. We surmise that beyond a certain threshold of toxin concentration, cellular activity is severely reduced and the cells’ sensitivity to changes in toxin stoichiometry is greatly diminished, resulting in this observation. Comparison with the results of Mckean reveals that HepG2 cells are more sensitive to mycotoxins [27]. For instance, the IC50 of AFB1 and T-2 on HepG2 cells was 1000 nM and 980 nM, respectively. Similarly, the IC50 of T-2 on human kidney epithelial cells was 154.6 nM [34]. These findings highlight that the same toxin can cause different degrees of inhibition of cellular activity in different cellular models.

In this study, the cytotoxic effects of AFB1 and T-2 on PK15 cells were investigated using different methods to determine if oxidative stress played a role. Oxidative stress, induced by the generation of ROS, is a major mechanism of neurotoxicity for several environmental and food contaminants, including the T-2 toxin, deoxynivalenol, and fumonisin B [35,36,37,38,39,40,41]. This process causes damage to DNA, promotes lipid peroxidation, protein damage, and ultimately leads to cell death. In the case of AFB1, lipid peroxidation occurs during its metabolic process in hepatocytes, activating the CYP450 enzyme system and resulting in the production of large amounts of ROS. The main consequence of ROS production by AFB1 is DNA damage [42,43]. Similarly, T-2 causes oxidative stress on cells, leading to decreased cellular activity, increased ROS levels, increased intracellular MDA levels, and increased activity of the antioxidant enzyme GSH-Px [44].

The results of this study showed that both ROS and MDA concentrations were significantly increased in the toxin group, with more significantly enhanced GSH-Px activity in the combined toxin group (*p* < 0.05), indicating that AFB1 and T-2 induce oxidative stress in PK15 cells and that the combined effect enhances oxidative stress in the cells. The study also revealed a non-significant increase in the amount of intracellular GSH in AFB1 and T-2 alone (*p* > 0.05), but a significant decrease in the amount of intracellular GSH in the combined toxin group (*p* < 0.05). This result is consistent with the results of OTA and AFB1 acting alone or in combination in 3D4/21 cells [29]. The decrease in GSH makes cells less able to resist oxidative damage, further increasing the level of ROS in the cell and leading to more severe cellular damage. Additionally, the relative expression levels of oxidative stress-related factors (GSH-Px, Hmox1, NQO1, and Keap1) were increased in all toxin groups, with more significant changes in the combined toxin group (*p* < 0.05). These changes in mRNA expression levels were similar to the corresponding changes in enzyme activity, further suggesting that the toxins may induce oxidative stress in PK15 cells.

In vivo, the study administered 2.5 mg/kg of AFB1, T-2, or a combination of the two (AT mixture) orally to Kunming mice for 14 days [45]. In this study, the determination of AFB1 and T-2 dosages was based on the oral LD50 of AFB1 and T-2, and set at 1/4 of that dosage. Additionally, our pre-experiments showed that when the dosage exceeded this level, mice in the T-2 and AFB1 groups could not survive until the end of the experiment [46,47]. During the experiment, we observed that both the T-2 and AT mice exhibited significantly lower final body weights compared to the other groups. Furthermore, the T-2 group had a lower body weight than the AT group. Our analysis suggests that this outcome was due to the minimal impact of the AFB1 toxin on body weight at the administered dose. In contrast, the T-2 toxin caused substantial damage to the mice’s digestive tracts, resulting in reduced weight gain or even weight loss. However, when both toxins were administered simultaneously, AFB1 interfered with the absorption of the T-2 toxin, reducing the severity of intestinal damage in the AT group compared to the T-2 group and consequently having a smaller effect on body weight. Of the four groups, only the AT group demonstrated a significant decrease in food intake. We hypothesize that this may be attributed to the higher concentration of toxin administered in the AT group (5 mg/kg) compared to the other groups, resulting in a greater impact on the mice’s appetite. Our results showed that, at equivalent doses, compared to the AFB1 group, mice in the T-2 group had lower food intake and body weight and exhibited a more significant increase in oxidative stress levels.

The toxicity of AFB1 and T-2 to mouse kidneys and their combined effect in an animal model of mouse oxidative stress were investigated using biochemical and oxidative stress-related indicators in kidney tissue. Histopathological analysis of kidney tissue sections showed the most severe damage in the AT group, while the T-2 group exhibited similar damage to the AT group, including glomerular atrophy and interstitial edema. Organ damage increases the permeability of cell membranes, leading to the release of intracellular enzymes into the blood, which results in increased enzyme activity in the serum [48,49]. Biochemical markers of the degree of organ damage were BUN, TC, and TG levels. In this study, ALP, BUN, TC, and TG activity were decreased, indicating severe renal damage. Previous studies have also demonstrated the toxic effects of AFB1 and AFM1 on the kidneys of mice, causing the abnormal expression of biochemical markers associated with renal function [50]. AFB1 and AFM1 induce kidney damage by increasing apoptosis and cell death [51]. Similarly, the toxicity of the T-2 toxin is linked to the production of ROS. The T-2 toxin causes an overproduction of ROS, disrupts cellular redox homeostasis, and results in kidney damage [52].

In this study, the combined action of AFB1 and T-2 increased MDA, decreased GSH, or did not change GSH-Px in the kidneys, indicating that the toxins accumulated as they were excreted from the kidneys into the body, causing some degree of oxidative damage to the kidneys and also resulting in antagonism after their combined action. The increase in Nrf2 levels in the kidney and the decrease in GSH-Px, Hmox1, and Keap1 levels suggest an imbalance in oxidative and antioxidant levels in the body and oxidative stress, consistent with the classical negative regulation of Nrf2 by Keap1 in mammals [53].

## 4. Conclusions

This study examined T-2/AFB1 toxicity and oxidative stress in PK15 cells and mouse kidneys. AFB1 and T-2 exhibited synergistic effects on PK15 cells, inducing oxidative damage and reducing cellular activity. Toxin treatment elevated ROS activity, MDA, GSH-Px, and oxidative stress-related mRNA expression in cells. In vivo, oral administration of the toxins caused severe histopathological changes in mice kidneys with elevated levels of ROS activity, MDA, GSH-Px, and mRNA expression. These results highlight the role of oxidative stress in AFB1 and T-2 toxicity, and their combined effects on public health. Further research is required to understand the underlying mechanisms and develop effective interventions.

## 5. Materials and Methods

### 5.1. Chemicals and Reagents

T-2 and AFB1 were purchased from Pribolab (Qingdao, China). The PK15 cells line was obtained from the Hunan Provincial Key Laboratory of Protein Engineering in Animal Vaccines, College of Veterinary Medicine, Hunan Agricultural University (Hunan, China). Dulbecco’s modified Eagle’s medium (DMEM), newborn calf serum (NBCS), penicillin-streptomycin Solution(P/S), trypsin-EDTA (0.25%), goat anti-rabbit secondary antibody, and goat anti-mouse secondary antibody were purchased from Thermofisher (Shanghai, China). Phosphate-buffered saline (PBS), thiazolyl blue tetrazolium bromide (MTT), Dimethyl sulfoxide (DMSO), Paraformaldehyde (4%), RNase-free ddH_2_O, and Radio Immunoprecipitation Assay Lysis Buffer (RIPA) were purchased from Solarbio (Beijing, China). NRF2 antibodies were purchased from Bioss Antibodies (Beijing, China). β-actin antibody was purchased from Cell Signaling Technology (Shanghai, China). Enhanced chemiluminescence (ECL) was purchased from Sigma Aldrich (St. Louis, MO, USA). SYBR green master mix was purchased from Vazyme (Nanjing, China).

### 5.2. Kits

Reactive oxygen species assay kit was purchased from Us Everbright Inc (Suzhou, China). Bicinchoninic acid protein quantification kit was purchased from Thermofisher (Shanghai, China). Lipid peroxidation MDA assay kit, GSH assay kit, and GSH-Px assay kit were purchased from Beyotime Biotechnology (Shanghai, China). M-MuLV first strand cDNA synthesis kit was purchased from Gene Copoeia Inc (GuangZhou, China).

### 5.3. Toxicity of T-2/AFB1 In Vitro

#### 5.3.1. Chemical Treatment

In order to conduct in vitro experiments on PK15 cells, a cell culture medium was prepared by adding 5% NBCS, 1% P/S, and DMEM. To obtain AFB1 and T-2 stock solutions, 1 mg of AFB1 or 10 mg of T-2 standards were dissolved in DMSO, respectively, and diluted with PBS to a final concentration of 10 mM. The AFB1 stock solution was further diluted to a working solution of 100 μM by adding 100 μL of stock solution to 9.9 mL of medium and stored at 4 °C. Similarly, the T-2 stock solution was diluted to a working solution of 10 μM by adding 20 μL of stock solution to 199.98 μL of medium and stored at 4 °C. All stock solutions were stored at −20 °C for long term storage.

MTT solution was prepared by fully dissolving MTT powder in PBS at a concentration of 5 mg/mL, followed by filtration through a 0.22 μm filter membrane to remove bacteria from the solution. The filtered MTT solution was dispensed into brown EP tubes for long-term storage at −20 °C. To prepare AFB1 and T-2 stock solutions for in vivo experiments, 10 mg of AFB1 or T-2 standards were dissolved in DMSO and diluted with sterile PBS to a final concentration of 1 mg/mL. These stock solutions were then used to administer AFB1 or T-2 to the mice orally.

#### 5.3.2. Chemical Treatment MTT Assay and Morphologic Evaluation

PK15 cells were seeded at a density of 4 × 10^3^ cells per well in 96-well culture plates and allowed to incubate for 24 h to ensure cell attachment. The cells were then treated with various concentrations of T-2 toxin (ranging from 0 to 10^3^ nM) and AFB1 toxin (ranging from 0 to 5 × 10^4^ nM) for 24 or 48 h [54,55]. Following the treatment, 10 μL of MTT solution (5 mg/mL) was added to each well and incubated for 4 h at 37 °C [56]. The culture medium was removed, and 150 μL of DMSO was added to each well to dissolve the formazan crystals. The absorbance value at 490 nm was measured using an enzyme marker after shaking the plate for 10 min under aseptic conditions. IC50 values were calculated using GraphPad Prism software (Version 8, Boston, MA, USA) to determine the concentration of each toxin that reduced cell viability by 50%. The IC50 values for AFB1 and T-2 were then used to estimate the combined IC50 value for both toxins.

For the assessment of cellular morphology, PK15 cells were seeded in 12-well culture plates at a density of 5 × 10^4^ cells per well and allowed to incubate for 24 h. The cells were then treated with AFB1 IC50, T-2 IC50, the estimated IC50 value for the combined action of both toxins (ATG), and the measured IC50 value for the combined action of both toxins (ATS). The cells were observed under a light microscope to evaluate changes in cellular morphology. The experimental groups included a blank control group, which did not receive any treatment, and four treatment groups. The blank control group was used to establish baseline cell morphology, and the treated groups were used to assess the effects of the toxins on cellular morphology.

#### 5.3.3. Measurement of ROS in Cell

To investigate the effect of T-2 and AFB1 on ROS levels in PK15 cells, a positive control sample was first prepared by diluting the positive control to a working concentration of 100 µM in double-free medium. PK15 cells were then treated with T-2 and AFB1 in 6-well culture plates for 6 h, and the positive control sample was added to cells and incubated at 37 °C for 30 min while shielded from light to increase ROS levels. After the reaction, the cells were washed twice with double-free medium to remove DCFH-DA that did not enter the cells. To measure ROS levels, the FL1 or BL1 channel of the flow cytometry was selected, and excitation at 488 nm and emission light at 530 nm were measured [57]. This method is valuable for assessing the oxidative stress caused by T-2 and AFB1 in PK15 cells, and the results provide important information on the cytotoxicity of these toxins.

#### 5.3.4. Measurement of Intracellular MDA in Cells

In order to assess the impact of T-2 and AFB1 on PK15 cells, the cells were treated with these toxins in 6-well plates for a period of 24 h. The culture medium was then removed and the cells were washed twice with PBS buffer, followed by lysis with cell lysate. The resulting supernatant was subjected to centrifugation at 12,000× *g* for 10 min at 4 °C and the protein concentration was determined for subsequent assays. To perform the MDA assay, 200 μL of MDA assay working solution was combined with 100 μL of PBS and standards or samples of varying concentrations, followed by mixing and heating in a boiling water bath for 15 min. After cooling to room temperature, the samples were centrifuged at 1000× *g* for 10 min, and 200 μL of the resulting supernatant was transferred to a 96-well plate for analysis. The amount of MDA present in each sample was then determined by measuring the absorbance at 532 nm using a microplate reader [58].

#### 5.3.5. Measurement of Intracellular GSH and GSH-Px in Cells

To assess the impact of T-2 and AFB1 at various doses on PK15 cells, the cells were treated with different concentrations of these mycotoxins for 24 h. The supernatants were then centrifuged at 4 °C, 10,000× *g* for 10 min and the level of glutathione (GSH) was measured using the Total Glutathione Assay Kit. Additionally, the supernatants were centrifuged at 4 °C, 12,000× *g* for 10 min, and the level of glutathione peroxidase (GSH-Px) was measured using the Total Glutathione Peroxidase Assay Kit [59]. These assays provide valuable insights into the antioxidant capacity of PK15 cells in response to different doses of T-2 and AFB1.

#### 5.3.6. Measurement of Relative Expression Levels of Intracellular Oxidative Stress-Related mRNAs

PK15 cells were first cultured for 24 h and then incubated with appropriate concentrations of AFB1 and T-2 for an additional 24 h. Total RNA was then extracted from the cells using Trizol reagent, and the extracted RNA was reverse transcribed into cDNA using the M-MuLV first strand cDNA synthesis kit. Primers were designed and the target sequence was sent to Changsha DynaScience Biotechnology. Table 4 displays the primer sequences for genes associated with oxidative stress. The resulting cDNA was then used as a template for the fluorescence quantitative PCR reaction, and the reaction system is detailed in Table 5. The reaction conditions were as follows: 95 °C for 10 min, 95 °C for 10 s, and 60 °C for 30 s for 39 cycles. The solubility curve was set at 95 °C for 15 s, 60 °C for 1 min, and 95 °C for 15 s. The relative expression of mRNA in the test group was determined by the 2^−ΔΔCt^ method, by comparing the Ct value of the test group gene with that of the internal reference gene. All PCRs were performed in triplicate.

### 5.4. Toxicity of T-2/AFB1 In Vivo

#### 5.4.1. Animal Experiments

Twenty KM mice, 50% male and 50% female, aged 30 days, were procured from Hunan SJA Laboratory Animal Co. They were distributed randomly into four groups, each containing five mice: the PBS group (NC group), AFB1 group (2.5 mg/kg·bw), T-2 group (2.5 mg/kg·bw), and the combined group: 2.5 mg/kg·bw AFB1 + 2.5 mg/kg·bw T-2 (AT group). The mice were orally administered the designated solutions once a day for a period of 14 days. All animal experiments were authorized by the Animal Ethics Committee of the Hunan Agricultural University, China. The mice were observed for signs of discomfort twice daily, and their body weights and the weight of the remaining feed were recorded.

#### 5.4.2. Serum Biochemistry Analyses

Three mice in each group were randomly selected for eye blood collection at 15 days of age, preserved in sodium heparin anticoagulant. Plasma was stored at 4 °C prior to analysis. After centrifugation, the supernatant was removed and individual plasma samples were tested for ALP, AST, ALT, TP, TC, TG, and BUN on a fully automated hematology analyzer (Perlong Medical, Nanjing, China).

#### 5.4.3. Observations of Kidneys Pathology

The kidney tissue samples were initially fixed in 4% paraformaldehyde for a minimum of 24 h and then paraffin-embedded. Subsequently, 4 µm sections were obtained from each paraffin-embedded tissue and mounted on slides [60]. These sections were then sequentially stained with hematoxylin and eosin and examined under a microscope for further analysis.

#### 5.4.4. Measurement of ROS in Kidneys of Mice

Tissue specimens were embedded in optimal cutting temperature compound (OCT) embedding agent and subsequently frozen for preservation. The embedded tissues were sectioned and carefully adhered to slides, which were subsequently stored in a refrigerator at a temperature of −80 °C. To ensure accurate and reliable results, tissue autofluorescence was quenched prior to staining with a ROS stain. Additionally, cell nuclei were re-stained with DAPI to enable clear visualization and accurate analysis. Finally, the slides were sealed to prevent any external contamination, and the sections were examined under a fluorescence microscope. High-quality images were collected for further analysis and interpretation.

#### 5.4.5. Measurement of MDA, GSH, and GSH-Px in the Kidneys of Mice

A portion of the kidney was homogenized with 10% saline, centrifuged at 4 °C, 2000 rpm for 10 min, and the supernatant was collected for measurement of MDA, GSH, and GSH-Px. The determination method was the same as in 5.3.

#### 5.4.6. Relative Expressions of Oxidative Stress-Related mRNAs in the Kidneys of Mice

The mouse kidney tissue was homogenized, and RNA was extracted using Trizol. The concentration of RNA was measured, and subsequently reverse transcribed to cDNA as described in Section 5.3.6. Primers were designed for the target sequences and submitted to Changsha Tsingke Biotechnology Co. for synthesis. The primer sequences for the oxidative stress-related genes are listed in Table 6.

### 5.5. Statistics

All experimental data were statistically analyzed using SPSS Statistics 26.0 software (SPSS Inc., Chicago, IL, USA). Fluorescence quantitative PCR test results were displayed using GraphPad Prism 8. One-way analysis of variance was used between groups, LSD test was used for two-way comparisons, tests were repeated three times, and test results are expressed as mean ± standard error, with *p* < 0.05 indicating a significant difference.

## Figures and Tables

**Figure 1 toxins-15-00503-f001:**
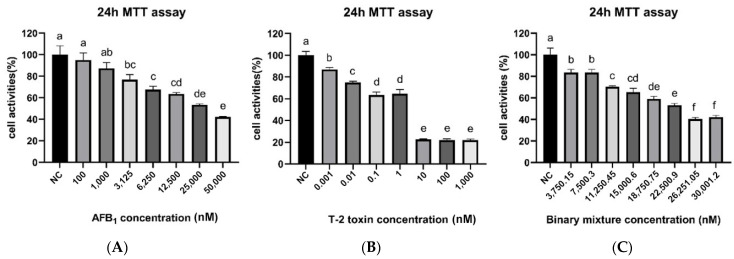
Cell activities (%) of PK15 treated with different concentrations of AFB1 for 24 h (**A**); Cell activities (%) of PK15 treated with different concentrations of T-2 for 24 h (**B**); Cell activities (%) of PK15 treated with an AFB1 and T-2 combination for 24 h (**C**). Values with the difference superscripts (a–f) are significant differences (*p* < 0.05).

**Figure 2 toxins-15-00503-f002:**
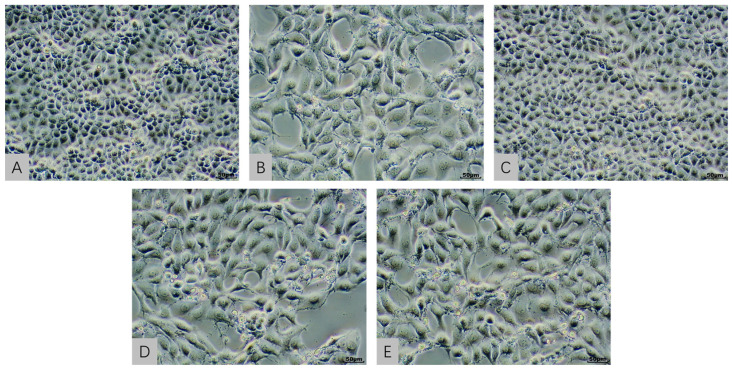
Micrograph of control cells (100×) (**A**); micrograph of 30,000-nMAFB1-treated cells (100×) (**B**); micrograph of 1.2-nM T-2-treated cells (100×) (**C**); micrograph of 30,000-nM AFB1- and 1.2-nM T-2-treated cells (100×) (**D**); micrograph of 23,272-nM AFB1- and 1-nM T-2-treated cells (100×) (**E**).

**Figure 3 toxins-15-00503-f003:**
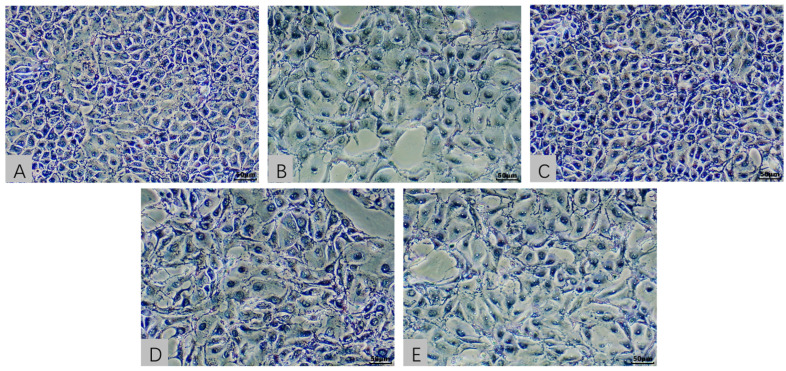
Micrograph after HE staining of control cells (100×) (**A**); micrograph after HE staining of 30,000-nMAFB1-treated cells (100×) (**B**); micrograph after HE staining of 1.2-nM T-2-treated cells (100×) (**C**); micrograph after HE staining of 30,000-nM AFB1- and 1.2-nM T-2-treated cells (100×) (**D**); micrograph after HE staining of 23,272-nM AFB1- and 1-nM T-2-treated cells (100×) (**E**).

**Figure 4 toxins-15-00503-f004:**
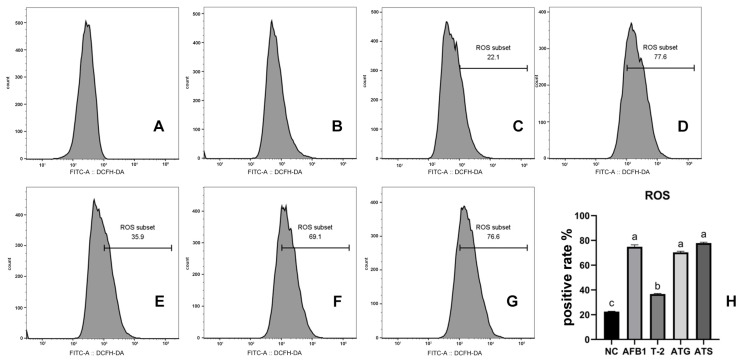
ROS production in T-2/AFB1-treated or untreated PK15 cells for 24 h. There are seven groups (**A**–**G**): negative control (**A**); positive control (**B**); cells were untreated (**C**); cell treated with AFB1 (**D**); cell treated with T-2 (**E**); cell treated with estimated IC50 of combined toxins (**F**); cell treated with observed IC50 of combined toxins (**G**); histogram of ROS positive rate in different groups (**H**). Values with the difference superscripts (a–c) are significant differences (*p* < 0.05). ATG means estimated IC50 group, ATS means observed IC50 group.

**Figure 5 toxins-15-00503-f005:**
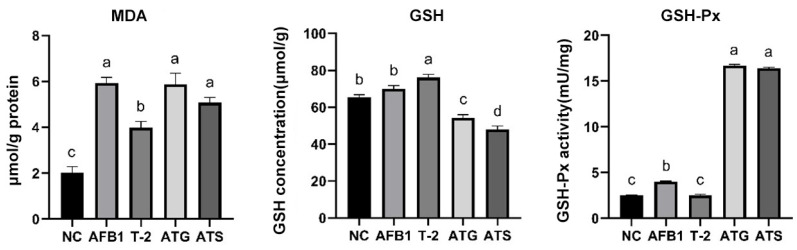
Comparison of MDA, GSH, and GSH-Px concentration in T-2/AFB1-treated or untreated PK15 cells. Values with the difference superscripts (a–d) are significant differences (*p* < 0.05). ATG means estimated IC50 group, ATS means observed IC50 group.

**Figure 6 toxins-15-00503-f006:**
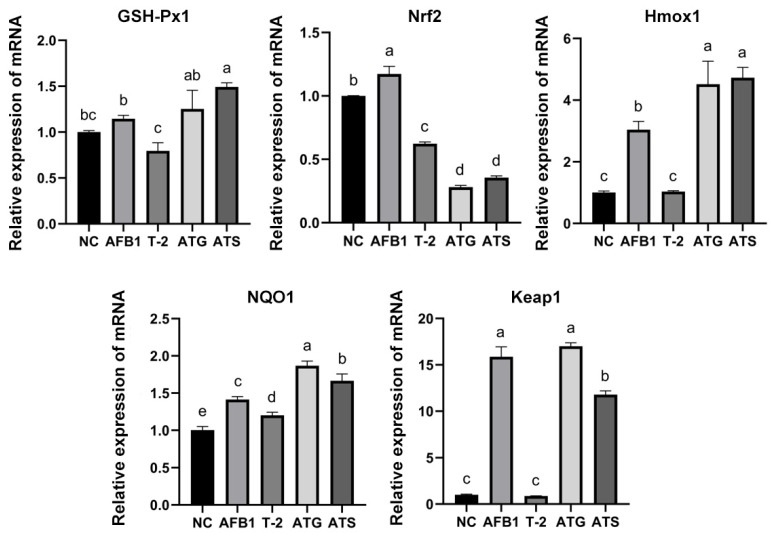
The relative expression of GSH-Px1, Nrf2, Hmox1, NQO1, and Keap1 mRNA in T-2/AFB1-treated or untreated PK15 cells. Values with the difference superscripts (a–e) are significant differences (*p* < 0.05). ATG means estimated IC50 group, ATS means observed IC50 group.

**Figure 7 toxins-15-00503-f007:**
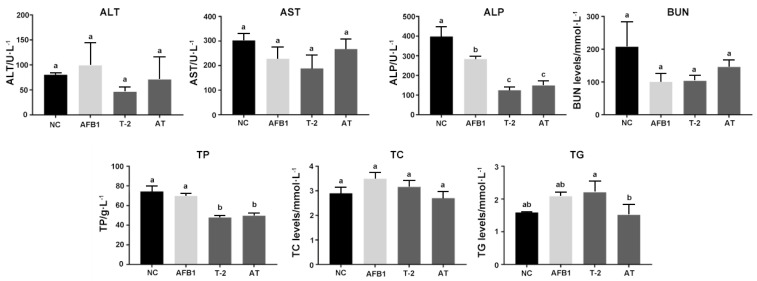
Changes of ALT, AST, ALP, BUN, TP, TC, and TG in T-2/AFB1-treated or untreated mice. Values with the difference superscripts (a–c) are significant differences (*p* < 0.05). AT means combined group.

**Figure 8 toxins-15-00503-f008:**
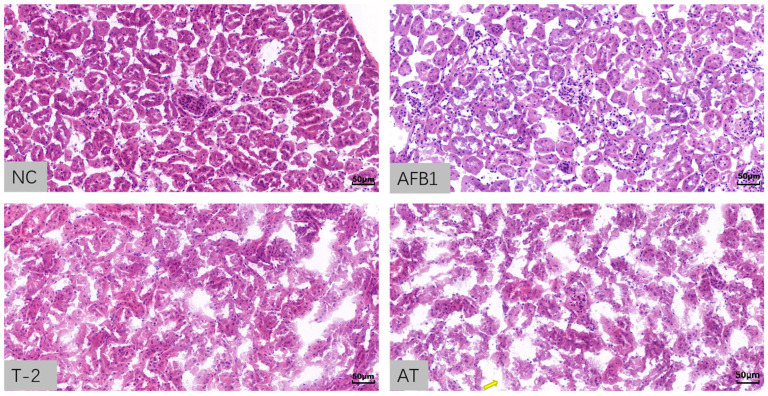
Micrograph of control kidney tissue in mice (200×) (NC); Micrograph of 2.5 mg/kg·bw AFB1-treated kidney tissue in mice (200×) (AFB1); Micrograph of 2.5 mg/kg·bw T-2-treated kidney tissue in mice (200×) (T-2); Micrograph of 2.5 mg/kg·bw AFB1 and T-2-treated kidney tissue in mice (AT). Arrows indicate widening of the luminal space of the cystic vesicle of Bowman’s capsule.

**Figure 9 toxins-15-00503-f009:**
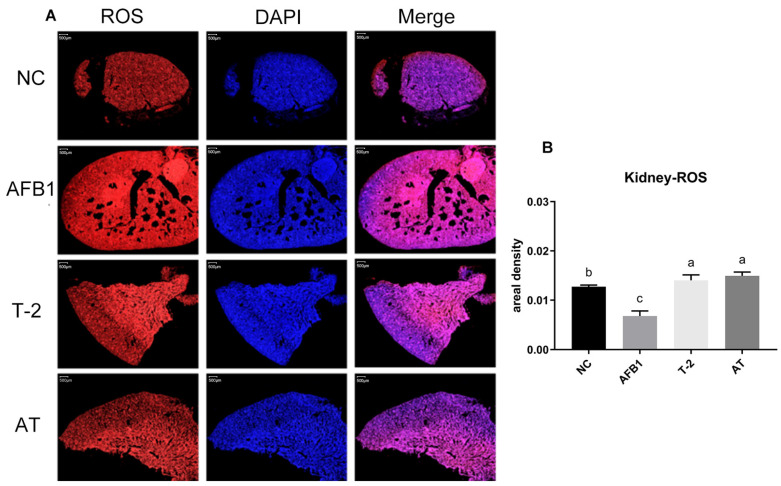
ROS content of mice kidney tissue. The microscope observation was 20×. (**A**): The level of ROS in the kidney tissue was detected by fluorescence microscope; (**B**): ROS quantitative analysis chart. Scale bar: 500 μm. Values with the difference superscripts (a–c) are significant differences (*p* < 0.05). AT means combined group.

**Figure 10 toxins-15-00503-f010:**
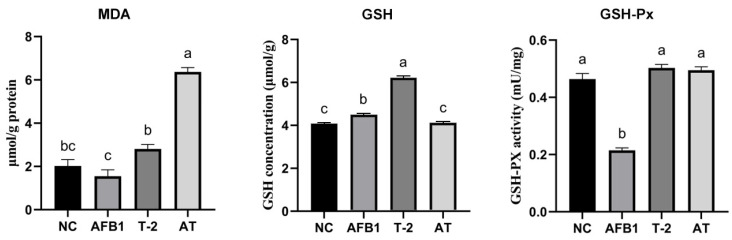
Comparison of MDA, GSH, and GSH-Px concentration in T-2/AFB1-treated or untreated kidney tissue of mice. Values with the difference superscripts (a–c) are significant differences (*p* < 0.05). AT means combined group.

**Figure 11 toxins-15-00503-f011:**
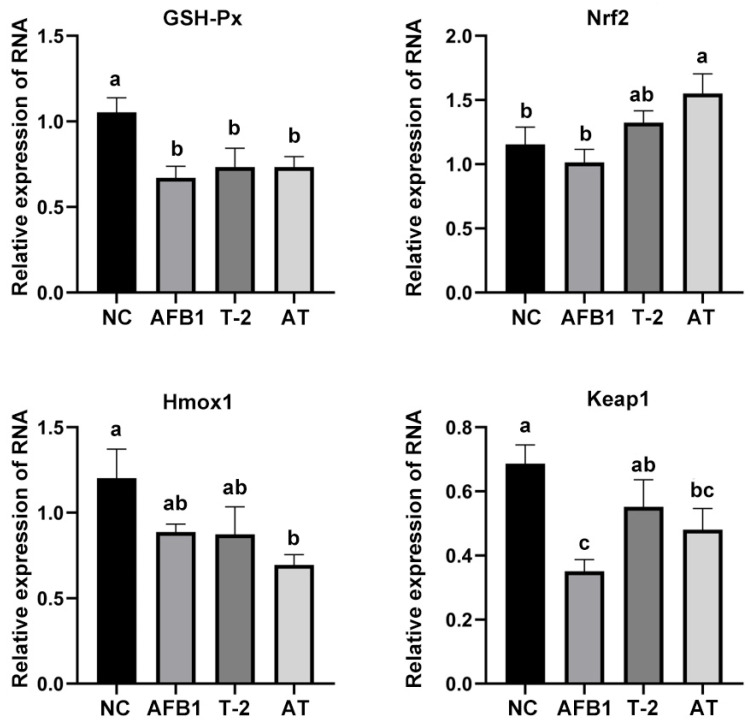
The relative expression of GSH-Px, Nrf2, Hmox1, and Keap1 mRNA in T-2/AFB1-treated or untreated kidney tissue of mice. Values with the difference superscripts (a–c) are significant differences (*p* < 0.05). AT means combined group.

**Table 1 toxins-15-00503-t001:** The IC50 result of AFB1 and T-2.

Cell	AFB_1_ IC_50_ (nM)	AFB_1_ 95% CL (nM)	T-2 IC_50_ (nM)	T-2 95% CL (nM)
PK15	30,060	22,480–43,740	1.179	0.5892–2.369

**Table 2 toxins-15-00503-t002:** Combinative toxicity of AFB1 and T-2 toxin in PK15 cells.

Concentration (IC_50_)	AFB_1_ (nM)	T-2 (nM)	Mix (nM)	Cell Viability (%) Mean ± SD
0 (NC)	0	0	0	100 ± 6
0.125	3750	0.15	3750.15	83 ± 3
0.25	7500	0.30	7500.30	83 ± 3
0.375	11,250	0.45	11,250.45	70 ± 1
0.50	15,000	0.60	15,000.60	65 ± 4
0.625	18,750	0.75	18,750.75	59 ± 2
0.75	22,500	0.90	22,500.90	53 ± 2
0.875	26,250	1.05	26,251.05	40 ± 1
1	30,000	1.20	30,001.20	42 ± 2

**Table 3 toxins-15-00503-t003:** Effects of mycotoxins on the weight and ADFI of mice.

Treatment	Initial Weight (g) Mean ± SD	Final Weight (g) Mean ± SD	ADFI (g) Mean ± SD
NC	25.51 ± 0.46	37.53 ± 0.89 ^a^	5.77 ± 0.35 ^a^
AFB_1_	26.06 ± 0.37	36.26 ± 1.20 ^a^	5.64 ± 0.55 ^a^
T-2	25.77 ± 0.35	20.50 ± 1.58 ^c^	4.81 ± 0.81 ^a^
AT	25.82 ± 0.32	26.47 ± 2.64 ^b^	3.20 ± 0.20 ^b^

Values with the difference superscripts (a–c) are significant differences (*p* < 0.05). Group NC: negative control group; Group AFB1: basal fed group treated with 2.5 mg/kg·bw AFB1; Group T-2: basal fed group treated with 2.5 mg/kg·bw T-2; Group AT: basal fed group treated with 2.5 mg/kg·bw AFB1 and T-2. ADFI means Average daily feed intake.

**Table 4 toxins-15-00503-t004:** Primer sequences of real-time quantitative PCR.

Gene	Upstream Primers (5′ → 3′)	Downstream Primers (5′ → 3′)
GAPDH	AGGGCATCCTGGGCTACACT	TCCACCACCCTGTTGCTGTA
GSH-Px1	CGTGCAACCAGTTTGGACAT	AGCATGAAGTTGGGCTCGAA
Nrf2	CACTAAACCCAATCCAACCCC	TTGTGAGATGAGCCTCCAAGC
Hmox1	CGCTCCCGAATGAACACTCT	GCGAGGGTCTCTGGTCCTTA
NQO1	GATCATACTGGCCCACTCCG	GAGCAGTCTCGGCAGGATAC
Keap1	CGGAGGCTATGATGGTCACA	ATTCCATCCCTAGCGTGCAG

**Table 5 toxins-15-00503-t005:** Real-time quantitative PCR system.

Reagent	Dosage
cDNA	1 μL
SYBR Green Mix	5 μL
Upstream primers	0.2 μL
Downstream primers	0.2 μL
RNase free H_2_O	3.6 μL

**Table 6 toxins-15-00503-t006:** Primer sequences of real-time quantitative PCR.

Gene	Upstream Primers (5′ → 3′)	Downstream Primers (5′ → 3′)
GAPDH	ATGACTCCACTCACGGCAAA	CGGCCTCACCCCATTTGATG
GSH-Px1	TGCAATCAGTTCGGACACCA	AAGGTAAAGAGCGGGTGAGC
Nrf2	ATCAGGCCCAGTCCCTCAAT	CAGCCAGCTGCTTGTTTTCG
Hmox1	CAGAAGAGGCTAAGACCGCC	CTCTGACGAAGTGACGCCAT
NQO1	GGTAGCGGCTCCATGTACTC	CCAGACGGTTTCCAGACGTT
Keap1	GATGGGCAGGACCAGTTGAA	CCGGGTCATAGCATTCCACA

## Data Availability

Not applicable.

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
