# Peer review of "Deciphering the Hazardous Effects of AFB1 and T-2 Toxins: Unveiling Toxicity and Oxidative Stress Mechanisms in PK15 Cells and Mouse Kidneys"

_toxins, 2023, doi:10.3390/toxins15080503_

Round 1
Reviewer 1 Report
The study highlights the importance of considering the combined effects of mycotoxins in animal feed, particularly AFB1 and T-2. This study aims to investigate the toxicity and oxidative stress of AFB1 and T-2 on Porcine Kidney-15 cell (PK15 cells) and mouse kidneys.
- Introduction very well writen and clear
-I suggest reformulate the aim
- Please padronize the manuscript using "mice" or "mouses"
- How is it possible to have no reference cited in the material and methods section?
- Between AFB1 and T2, alone, which one is shown to be the most toxic? I know the aim of the study was to evaluate the combined effects, but once you made the experiments also separately, It's important to talk about this
- What about the discussion on the differences in the body weights in the mice? And about the results after the evaluation of signs of discomfort on them?
- 87: and between 0.001-0.1 nM or 1-10 nM?
- 127: What do you think happened for the estimated concentration had a minimum effect for toxin T2?
- 397 - 398: How were the concentrations chosen? Why you didn't choose the same range for both compounds?
- 279: Digitation mistake?
- 279: I think is better to put this last sentence in the next paragraph, when you write about the combined action of mycotoxins
- Please, when possible, put all the concentrations in the same unit (or everything in nM, or everything in μM, for example)
- 302: please space between B and [32-38]
- 325: please space between days and [42]
Author Response
Title: Deciphering the Hazardous Effects of AFB1 and T-2 Toxins: Unveiling Toxicity and Oxidative Stress Mechanisms in PK15 Cells and Mouse Kidney
Authors: Lingchen Yang*, Shuai Xiao, Yingxin Wu, Suisui Gao, Mingxia Zhou, Zhiwei Liu, Qianbo Xiong, Lihuang Jiang, Guoxiang Yuan, Linfeng Li
Manuscript ID: toxins-2502659
Dear Editor and Reviewers,
Thank you very much for your valuable comments and suggestions on our manuscript. Following the reviewers’ comments, we have modified and improved our manuscript according to your kind advices and referee’s detailed suggestions. Enclosed please find the responses to the referees. We sincerely hope this manuscript will be acceptable to be published on Toxins.
Reply to Reviewers
We thank the reviewer for his/her constructive criticisms that have helped us to improve our manuscript. The point-by-point response to the comments is given below.
Thank you very much for all your help and looking forward to hearing from you soon.
Best regards
Sincerely yours
Prof. Yang
Please find the following Response to the comments of referees:
Question 1: I suggest reformulate the aim.
Response to Question 1: We are deeply appreciative of the reviewers for their valuable suggestions. In response, we have updated the objective of our research in both the Abstract and Introduction sections. (Line 15-17 and Line 88-90)
Question 2: Please padronize the manuscript using "mice" or "mouses"
Response to Question 2: Thank you for your recommendation. I have carefully reviewed the entire manuscript and have consistently used “mice” as the plural form of “mouse”.
Question 3: How is it possible to have no reference cited in the material and methods section?
Response to Question 3: We would like to express our sincere gratitude to the reviewers for their valuable suggestions. In response, we have incorporated additional references [51-57] into this section.
Question 4: Between AFB1 and T2, alone, which one is shown to be the most toxic? I know the aim of the study was to evaluate the combined effects, but once you made the experiments also separately, It's important to talk about this.
Response to Question 4: We are deeply appreciative of the reviewers for their valuable suggestions. The severity of cellular and animal damage caused by mycotoxins is dependent on the concentration of the exposed toxin. In our cellular experiments, we evaluated the toxicity of AFB1 and T-2 toxins, both individually and in combination, by treating PK15 cells with varying concentrations of each toxin. Our initial findings revealed that, at equivalent doses, cell viability in the T-2 group was significantly lower than in the AFB1 group (Figure 1A and B), indicating that T-2 had a more pronounced effect on cell viability at the same concentration. Additionally, through this experiment, we determined the IC50 of T-2 to be 1.179 nM (subsequently used at 1.2 nM) and the IC50 of AFB1 to be 30.06 μM (subsequently used at 30 μM). To investigate the combined toxicity of the two toxins, we designed subsequent cellular experiments based on the IC50 values of both toxins. As a result, we observed that, at these concentrations, AFB1 was more toxic than T-2.
In our in vivo experiments, we referred to literature reports that the oral LD50 of AFB1 for mice is 9mg/kg body weight1 and that of T-2 is 9.6-10.5mg/kg body weight2. Therefore, we set the toxin dosage for both AFB1 and T-2 at 1/4 of the LD50, or 2.5 mg/kg. Additionally, based on our pre-experimental results, we ultimately decided to use the same dose (2.5mg) for both toxins in subsequent in vivo toxicity studies.
Our results showed that, at equivalent doses, compared to the AFB1 group, mice in the T-2 group had lower food intake and body weight and exhibited a more significant increase in oxidative stress levels. In summary, when exposed to equivalent doses of both toxins, T-2 is more toxic than AFB1.
Question 5: What about the discussion on the differences in the body weights in the mice? And about the results after the evaluation of signs of discomfort on them?
Response to Question 5: We would like to express our sincere gratitude to the reviewers for their insightful suggestions. During our experiment, we observed that both the T-2 and AT mice exhibited significantly lower final body weights compared to the other groups, with the T-2 group having a lower body weight than the AT group. Our analysis indicates that this result was due to the negligible impact of AFB1 toxin on body weight at the administered dose. In contrast, the T-2 toxin inflicted considerable damage to the mice’s digestive tract, leading to diminished weight gain or even weight loss. However, when both toxins were co-administered, AFB1 impeded the absorption of T-2 toxin, mitigating the severity of intestinal damage in the AT group compared to the T-2 group and thus exerting a smaller effect on body weight. Among the four groups, only the AT group exhibited a significant reduction in food intake. We postulate that this may be due to the higher concentration of toxin administered in the AT group (5 mg/kg) compared to the other groups, resulting in a more pronounced impact on the mice’s appetite. We have incorporated this explanation into the discussion section (Line 371-385). Given the opportunity, we plan to design and conduct further related experiments in the future.
Question 6: What about the discussion between 0.001-0.1 nM or 1-10 nM?
Response to Question 6: We would like to express our sincere gratitude to the reviewers for their valuable suggestions. In response, we have included a description of the 0.001-0.1 nM range (lines 101-102) and have also provided a description of the 1-10 nM range (lines 103).
Question 7: What do you think happened for the estimated concentration had a minimum effect for toxin T2?
Response to Question 7: We are very grateful to the reviewers for their suggestions. Based on our observations, at this concentration of T-2, cellular activity appears to be diminished and no significant differences in cell morphology are evident compared to the NC group.
Question 8: How were the concentrations chosen? Why you didn't choose the same range for both compounds?
Response to Question 8: We are deeply appreciative of the reviewers for their valuable suggestions. This concentration was determined by reviewing relevant literature and conducting preliminary experiments (Line 95-105). We selected the IC50 values of both toxins for subsequent toxicity experiments for two main reasons. Firstly, we observed that, at equivalent toxin concentrations, T-2 had a significantly greater impact on cells than AFB1. As shown in Figure 1A and B, when the toxin concentration was 100 nM, AFB1 had no significant effect on cell viability, whereas T-2 caused a significant decrease in cell viability to only about 20%. If we were to conduct a combined experiment using the same concentration for both toxins, it might not accurately reflect the impact of AFB1 on cells. Secondly, we chose to use the IC50 values as the basis for selecting concentrations because it allows us to more comprehensively evaluate the combined effects of AFB1 and T-2 and to further analyze whether they exhibit synergistic, antagonistic, or other interactions (Line 300-302).
Question 9: I think is better to put this last sentence in the next paragraph, when you write about the combined action of mycotoxins
- Please, when possible, put all the concentrations in the same unit (or everything in nM, or everything in μM, for example)
Response to Question 9: We are very grateful to the reviewers for their suggestions. Changes have been made as required (Line 388-390).
Question 10: Please space between B and [32-38]. space between days and [42]
Response to Question 10: We are very grateful to the reviewers for their suggestions. Changes have been made as required (Line 344 and Line 367).
References:
- Gugliandolo, E.; Peritore, A. F.; D'Amico, R.; Licata, P.; Crupi, R., Evaluation of Neuroprotective Effects of Quercetin against Aflatoxin B1-Intoxicated Mice. Animals (Basel) 2020,10(5).
- Kalantari, H.; Moosavi, M., Review on T-2 Toxin. Jundishapur Journal of Natural Pharmaceutical Products 2010,5(1), 26-38.
Reviewer 2 Report
This manuscript describes the combined impacts of AFB1, and T-2 on the Porcine Kidney-15 cell line (PK15). The cells were treated with the different AFB1 and T-2 concentrations for 24 h, and the cell activity, morphological evaluation, and oxidative stress-related indicators were examined. This study highlights the importance of considering the combined effects of mycotoxins in animal feed, particularly AFB1 and T-2. This manuscript is well-written, and the protocols are precisely described. The findings have important implications for animal feed safety and represent a base for further investigation of the combined effects of mycotoxins.
However, I would recommend a major revision:
* Some typographical errors are recommended to be corrected for greater clarity.
* Additional information about the AFB1 is recommended to be added to the manuscript.
* Some additional explanation is recommended to be added to the manuscript for a better understanding of certain results.
Introduction:
Page 1:
18: Please italicize Aspergillus flavus strain and Aspergillus niger.
25-33: Please add information about the AFB1, and mention that AFB1 is a procarcinogen and belongs to group 1 of carcinogens. Quote the papers:
Chem. Res . Toxicol. 2014; 27(12):2136-2147
Foods 2021, 10, 1331
Results
Page 2:
84: Please rewrite “50μM” to “50 μM”.
94: Please rewrite “ofPK15” to “of PK15”.
Page 3:
Figure 1B: Please explain possible reasons why the cell activity is higher at 1 nM than at 0.1 nM.
114-116: Please choose how to write numbers 15,000.6 nM, and 23,273 nM, since writing it differently than before (23273 nM) can cause confusion.
Table 2: Please add an explanation of the results, why at 3750.15, the cell activity is the same as at 7500.3. Furthermore, why is it higher at 30001.2 than at 26251.05?
Page 5:
144: Rewrite “pk15” to PK15”
155: Delete “in” to make the sentence clearer.
Discussion:
Page 11
281: Please add more examples of cytotoxicity tests on other cell lines, such as HepG2. Quote the paper:
Biosensors 2022, 12, 160
Materials and methods:
Page 12
378, 394: The first letter of the title of the chapter should be written in capital.
Ok.
Author Response
Title: Deciphering the Hazardous Effects of AFB1 and T-2 Toxins: Unveiling Toxicity and Oxidative Stress Mechanisms in PK15 Cells and Mouse Kidney
Authors: Lingchen Yang*, Shuai Xiao, Yingxin Wu, Suisui Gao, Mingxia Zhou, Zhiwei Liu, Qianbo Xiong, Lihuang Jiang, Guoxiang Yuan, Linfeng Li
Manuscript ID: toxins-2502659
Dear Editor and Reviewers,
Thank you very much for your valuable comments and suggestions on our manuscript. Following the reviewers’ comments, we have modified and improved our manuscript according to your kind advices and referee’s detailed suggestions. Enclosed please find the responses to the referees. We sincerely hope this manuscript will be acceptable to be published on Toxins.
Reply to Reviewers
We thank the reviewer for his/her constructive criticisms that have helped us to improve our manuscript. The point-by-point response to the comments is given below.
Thank you very much for all your help and looking forward to hearing from you soon.
Best regards
Sincerely yours
Prof. Yang
Please find the following Response to the comments of referees:
Question 1: Please italicize Aspergillus flavus strain and Aspergillus niger.
Response to Question 1: We are very grateful to the reviewers for their suggestions. Changes have been made as required (Line 38).
Question 2: Please add information about the AFB1, and mention that AFB1 is a procarcinogen and belongs to group 1 of carcinogens.
Response to Question 2: We are very grateful to the reviewers for their suggestions. We have added the information as suggested (Line 44-47).
Question 3: Please rewrite “50μM” to “50 μM”.
Response to Question 3: We are deeply appreciative of your advice. We've made changes and standardized all concentrations to the same unit (nM) (Line 97-98).
Question 4: Please rewrite “ofPK15” to “of PK15”.
Response to Question 4: We sincerely apologize for this oversight and have made the necessary revisions as requested (Line 113).
Question 5: Please explain possible reasons why the cell activity is higher at 1 nM than at 0.1 nM.
Response to Question 5: We appreciate your inquiry. Our observations indicate that cell viability was marginally higher when the T-2 toxin concentration was 1 nM compared to 0.1 nM, but the difference was not statistically significant. We infer that as the toxin concentration increases, cell activity decreases and the sensitivity of the cells to the toxin diminishes. When the T-2 toxin concentration is within the 0.1-1 nM range, the sensitivity of PK15 cells to it remains constant, resulting in no significant difference in cell activity between the two concentrations.
Question 6: Please choose how to write numbers 15,000.6 nM, and 23,273 nM, since writing it differently than before (23273 nM) can cause confusion.
Response to Question 6: We are very grateful to the reviewers for their suggestions. In response, we have made the necessary revisions to the figures throughout the manuscript.
Question 7: Please add an explanation of the results, why at 3750.15, the cell activity is the same as at 7500.3. Furthermore, why is it higher at 30001.2 than at 26251.05?
Response to Question 7: At concentrations of 3750.15 and 7500.3 nM, the toxin dosage was low, allowing the cells to withstand the toxin to some degree. Consequently, there was a reduction in cell viability at these two concentrations, but no significant difference between them. Cell viability decreased as the toxin concentration increased, but was slightly higher at 30,001.2 nM than at 26,251.05 nM. We surmise that beyond a certain threshold of toxin concentration, cellular activity is severely reduced and the cells’ sensitivity to changes in toxin stoichiometry is greatly diminished, resulting in this observation.
Question 8: Rewrite “pk15” to PK15”
Response to Question 8: We sincerely apologize for this oversight and have made the necessary revisions as requested (Line 165).
Question 9: Delete “in” to make the sentence clearer.
Response to Question 9: We sincerely apologize for this oversight and have made the necessary revisions as requested (Line 175).
Question 10: Please add more examples of cytotoxicity tests on other cell lines, such as HepG2.
Response to Question 10: We are deeply appreciative of the reviewers for their valuable suggestions. We have I've added more examples at the suggestions (Line 313-318).
Question 11: 378, 394: The first letter of the title of the chapter should be written in capital.
Response to Question 11: We sincerely apologize for this oversight and have made the necessary revisions as requested (Line 440 456).
Round 2
Reviewer 2 Report
The authors successfully resolved all issues raised by this Reviewer. Consequently, the manuscript has been significantly improved and can be in its current version recommended for publication in Toxins.
OK.